# Determining the impact of an artificial intelligence tool on the management of pulmonary nodules detected incidentally on CT (DOLCE) study protocol: a prospective, non-interventional multicentre UK study

Emma O'Dowd [1], Marko Berovic,[2] Matthew Callister,[3] Christos V Chalitsios,[4] Disha Chopra,[5] Indrajeet Das,[6] Adrian Draper,[7] Justin L Garner,[8] Fergus Gleeson,[9] Sam Janes,[10] Martyn Kennedy,[3] Richard Lee,[11] Fabrizio Mauri,[5] Tricia M McKeever [4], William McNulty,[2] James Murray,[12] Arjun Nair,[13] John Park,[9] Janette Rawlinson,[14] Gurdeep Singh Sagoo [15], Andrew Scarsbrook,[3] Pallav Shah,[16] Rajini Sudhir,[6] Ambika Talwar,[9] Ricky Thakrar,[17] Johnathan Watkins,[5] David R Baldwin[1]

For numbered affiliations see end of article.

**Correspondence to**
Dr Emma O'Dowd;
emma.o'dowd@nottingham.ac.uk

## ABSTRACT

**Introduction** In a small percentage of patients, pulmonary nodules found on CT scans are early lung cancers. Lung cancer detected at an early stage has a much better prognosis. The British Thoracic Society guideline on managing pulmonary nodules recommends using multivariable malignancy risk prediction models to assist in management. While these guidelines seem to be effective in clinical practice, recent data suggest that artificial intelligence (AI)-based malignant-nodule prediction solutions might outperform existing models.

**Methods and analysis** This study is a prospective, observational multicentre study to assess the clinical utility of an AI-assisted CT-based lung cancer prediction tool (LCP) for managing incidental solid and part solid pulmonary nodule patients vs standard care. Two thousand patients will be recruited from 12 different UK hospitals. The primary outcome is the difference between standard care and LCP-guided care in terms of the rate of benign nodules and patients with cancer discharged straight after the assessment of the baseline CT scan. Secondary outcomes investigate adherence to clinical guidelines, other measures of changes to clinical management, patient outcomes and cost-effectiveness.

**Ethics and dissemination** This study has been reviewed and given a favourable opinion by the South Central—Oxford C Research Ethics Committee in UK (REC reference number: 22/SC/0142).

Study results will be available publicly following peer-reviewed publication in open-access journals. A patient and public involvement group workshop is planned before the study results are available to discuss best methods to disseminate the results. Study results will also be fed back to participating organisations to inform training and procurement activities.

## STRENGTHS AND LIMITATIONS OF THIS STUDY

⇒ This is a prospective study examining the use of artificial intelligence (AI)-based computer-aided diagnosis (CADx) for pulmonary nodules.
⇒ This study will examine the clinical utility of an AI-based CADx tool for pulmonary nodule management.
⇒ The study is non-interventional in nature.
⇒ The analysis of some of the secondary objectives is exploratory due to potential sample size restrictions.

**Trial registration number** NCT05389774.

## INTRODUCTION

From 2017 to 2019, lung cancer caused 34 771 deaths in the UK per annum, more than prostate and bowel cancer combined (28 846), the next most common causes of cancer death. Over the same period, there were 24% more deaths in women from lung cancer than from breast malignancy (14 140 vs 11 415).[1] This is because almost three-quarters of people with lung cancer present with stage III or IV disease when the prognosis is poor despite recent advances in systemic anti-cancer treatment. Lung cancer detected at an early stage, usually by thoracic CT, has a much better prognosis. Screening with low-dose CT thorax has recently been confirmed to reduce lung cancer and all-cause mortality.[2–4] Central to the success of screening is the accurate management of pulmonary nodules, which

may represent early-stage lung cancer, but are far more often benign. Pulmonary nodules are also a common finding on CT scans done for other reasons. In some settings, this may yield more early lung cancers than screening. Identifying malignant pulmonary nodules on CT scans is the primary way to detect lung cancer at stage I, when the prognosis is best. By definition, isolated pulmonary nodules without other lung changes are stage I if proven to be lung cancer. Currently, most nodules are detected outside screening programmes in the UK, providing an important rationale for this study to address the effective management of these incidentally detected nodules.

The British Thoracic Society (BTS) guideline on managing pulmonary nodules was published in 2015.[5] This guidance was the first to recommend semiautomated volumetry as the standard for nodule measurement and the use of multivariable malignancy risk prediction models to assist in management. While these guidelines seem to be effective in clinical practice,[6] recent data suggest that artificial intelligence (AI)-based malignant-nodule prediction solutions appear to outperform existing multivariable models.[7 8] This could result in a considerable reduction in the number of follow-up CT, which is much needed in the National Health Service (NHS), where current and forecasted demand for radiology services exceeds capacity.[9 10]

Optellum has developed and extensively validated an AI-based computer-aided diagnostic (CADx) tool for lung cancer prediction (LCP) in pulmonary nodules, which has UK Conformity Assessed marking under UK medical device regulations. The AI model underpinning the LCP is a convolutional neural network (CNN), referred to hereafter as the 'LCP-CNN'. LCP-CNN was derived and internally validated using data from diverse populations from the USA and UK. The model was then externally validated in two independent cohorts of patients with incidentally detected pulmonary nodules, amounting to over 1500 patients in aggregate.[7 8] The LCP-CNN demonstrated greater accuracy in these studies than the Brock and Mayo malignancy risk models (used in managing incidentally detected nodules in the UK and USA, respectively). The product is provided within the 'Virtual Nodule Clinic' (VNC), a software device used in the tracking, assessment and characterisation of incidentally detected pulmonary nodules.

The study aims to assess this new AI tool in a prospective real-world setting in the UK. This will be done by calculating malignancy risk using LCP and measuring how management would differ if it was governed by the risk provided. The aim is to provide evidence as to whether LCP should be the new standard of care in pulmonary nodule management.

## METHODS AND ANALYSIS

The study acronym is 'DOLCE' (Determining the impact of Optellum's LCP artificial intelligence solution on service utilisation, health Economics and patient outcomes).

### Design and setting

DOLCE is a multicentre prospective observational cohort study recruiting patients with 5–30 mm solid and part-solid pulmonary nodules incidentally detected on thoracic CT scans performed as part of routine practice. The study aims to recruit 2000 patients from 12 centres.

Patients will be asked to consent to and be enrolled at twelve acute NHS hospital trusts in England. A recruitment period of 1 year is planned, with 1 year of follow-up to collect data and ascertain the ground-truth diagnosis of recruited patients.

The start date was March 2023 with the planned recruitment end date being February 2024. With 1 year of follow-up, this will mean a planned end date of February 2025.

### Clinical pathway

This is a non-interventional study. There will be no anticipated change to the standard of care the recruited patients receive throughout the study's duration. The potential impact of AI-modified management will be assessed by comparing routine clinical management with that which would have been conditional on the LCP risk score. The scores will be provided to clinicians after routine management decisions have been made, either during the same multi-disciplinary team (MDT) meeting or at a separate research meeting. A hypothetical clinical decision based on the LCP score will then be recorded.

### Study population

Patients who have undergone CT that has detected a pulmonary nodule and was performed for a reason other than suspicion of lung cancer will be reviewed for study eligibility (box 1).

### AI model

The output of the LCP is a risk score that is an integer from 1 to 10 and forms part of Optellum's VNC patient management software. A score of 1 indicates the nodule is highly likely to be benign, while a score of 10 is most likely to be malignant. The AI model underpinning the LCP is the LCP-CNN, the raw score available to researchers as a continuous value from 0 to 100, as previously reported. LCP-CNN was derived and internally validated using data from diverse populations from the USA and UK. The model was then externally validated in two independent cohorts of patients with incidentally detected pulmonary nodules, amounting to over 1500 patients in aggregate.[7 8]

In this study, the LCP will be compared with standard care in three ways:

1. LCP as an integer (1–10) risk score to aid physician decision-making (physician guided by AI score).
2. LCP-CNN alone as a threshold-based rule-out recommendation (AI binary decision to discharge or not).
3. LCP-CNN combined with a threshold-based rule-out recommendation to aid physician decision-making

## Box 1  Patient eligibility criteria

### Inclusion criteria

Patients aged 35 years or over.

Patients with a baseline CT study with at least one incidentally detected solid or part-solid pulmonary nodule that is 5–30 mm in maximum axial diameter for the whole lesion, measured using manual electronic callipers, and that is not fully calcified. The solid component of the part-solid nodule must be ≥80% of the total nodule size.

Patients with a baseline CT study which includes at least one series with acquisition parameters which meet Virtual Nodule Clinic requirements.

The patient has a baseline CT study that includes at least one series that comprises at least one full-inspiration breath-hold scan without a high degree of contrast media and does not exhibit quality issues (eg, motion artefacts).

### Exclusion criteria

Patients who received a diagnosis of cancer in the last 5 years.

Patients who have thoracic implants that adversely impact the imaging appearances of the nodule.

Patients who have more than five reported pulmonary nodules of any size or type, excluding fully calcified nodules (a marker of prior granulomatous infection) or metastatic lung cancer.

Patients who have one or more additional nodules that are already undergoing follow-up according to pulmonary nodule management standard care.

Patients who have one or more additional nodules that are pure ground glass opacity of 5 mm in maximum axial diameter for the whole lesion measured using manual electronic callipers.

Patients who have one or more additional nodules >30 mm in maximum axial diameter for the whole lesion measured using manual electronic callipers.

Patients with part-solid nodules where the solid component is <80% of the total nodule size.

(physician guided by AI with binary recommendation to discharge or not).

### Data collection

Study teams at the 12 NHS Trusts from which patients will be recruited will identify eligible patients and obtain informed consent. Consent will be obtained verbally (for patients who would not normally come on-site as part of standard care) or via written consent. Consent will be sought to analyse standard-care CT scans, associated clinical meta-data collection and the health economics assessment.

Once a patient has given consent and enrolled into the study, their baseline CT scan will be sent to VNC. Authorised research staff will then log into VNC, select the nodules of interest and record the LCP score for that nodule. The local standard nodule service decision-makers will provide a standard-care decision for eligible, consented patients via their usual nodule management pathway. The standard-care decision for eligible consented patients will then be recorded in an electronic case report form (eCRF) without knowing the LCP score.

To assess LCP under scenario #1 as described under AI model, the clinical team will thereafter be given the LCP score and relevant patient data in the nodule MDT or as part of a separate research meeting. At this point, the clinical team should remain blinded to the outcome of any actual-care investigation (eg, positron emission tomography (PET)/CT scan, biopsy). Their decision on the next management step, had they been guided by the LCP score, will then be recorded in the eCRF. The clinical team will have the option to modify management at this point. Any modification of management will be recorded in the eCRF.

Subsequent follow-up images for each patient will be sent to the VNC over the course of their nodule follow-up, and the final nodule diagnosis will be recorded. Since standard patient care can sometimes deviate from BTS guidelines, all participating sites will also collect information on patient and nodule characteristics necessary to follow the BTS guidelines. In this way, AI-guided management can be compared with BTS-compliant management. Other data points necessary to evaluate service utilisation (eg, additional scans, biopsies and treatments for lung nodules or lung cancers) will also be recorded in the eCRF.

In line with Standard Protocol Items: Recommendations for Interventional Trials - Artificial Intelligence (SPIRIT-AI) guidelines, performance errors that would have happened if the LCP or LCP-guided decision-making were part of standard care will be analysed and reported as outcome measures for the primary objective of this study.

Evaluation of LCP as threshold-based tests (scenarios #2 and #3 as described under AI model), which require access to LCP-CNN values, will be performed as a batch at the end of the study follow-up period. The threshold will be defined prospectively based on values from training datasets of similar patient populations. The use of the recommendation with physicians making the decision will be evaluated via a separate research meeting at the end of the 12-month follow-up period.

For cost-effectiveness modelling, quality of life data will be collected by inviting patients to complete an EQ-5D-5L quality of life questionnaire at two time points: (a) as close as possible to the baseline CT scan on which their pulmonary nodule was detected and (b) at a 1-year time point afterwards. A participant diary about primary care usage will also be collected at the 1-year mark.

### Ground truth diagnosis

Patients will be classified into whether they have benign or malignant nodules as follows:

1. Benign-nodule patients:
   i. By resolution or stability of all qualifying incidental pulmonary nodules as determined from volumetry-based measurement on imaging follow-up over 12 months.
   ii. By histopathology from biopsy or surgery of all qualifying incidental pulmonary nodules (within 12 months of baseline CT scan).
2. Malignant-nodule patients:

i. By histopathology from biopsy or surgery (within 12 months of baseline CT scan) of at least one qualifying incidental pulmonary nodule.

3. Probable benign-nodule patients:

i. By resolution, stability or shrinkage of qualifying incidental pulmonary nodules that are not in ground-truth outcomes #1 or #2 above as determined from volumetry-based measurement on imaging follow-up conducted between 3 and 12 months after the baseline CT scan.

ii. By resolution, stability or shrinkage of qualifying incidental pulmonary nodules that are not in ground-truth outcomes 1 or 2 above as determined from diameter-based measurement on imaging follow-up conducted 12 months after the baseline CT scan (supplemented with a volume-based measurement of the nodule that will be performed for research purposes).

4. Probable malignant-nodule patients:

i. By clinical diagnosis of lung cancer, in whom a biopsy or resection is not possible, based on high clinical probability and who are registered at a lung cancer MDT/and/or receive non-surgical treatment (stereotactic ablative radiotherapy (SABR)/radiotherapy/microwave ablation).

Volumetry-based measurement of nodule stability—and hence a benign diagnosis—will be satisfied if the lung nodule volume on the follow-up CT scan was <125% of the lung nodule volume on the baseline CT scan.

If the standard care at a particular site does not incorporate volumetry-based measurements, the volumetric measurement will be applied as part of the research study to the relevant follow-up CT scans.

Any lung nodule patients who do not fall into either of the two ground-truth outcomes 1 or 2 above will be considered to have an indeterminate diagnosis at the time of analysis and will be analysed with and without the inclusion of the ground-truth outcomes in 3 and 4.

### Outcome measures

The outcome measures for the primary and secondary objectives are outlined in table 1.

### Statistical analysis

#### Sample size

For calculations relating to the primary objectives of a measured difference between standard care and LCP for:

▶ Number and percentage of cancer patients discharged: results from thresholds defined for discharge in a previous study are used.[7] Based on previous data within the cancer group, assuming for the discordant pairs that the Brock model threshold (representing the standard of care) identified 0.4% of cancers for which the LCP threshold would have recommended discharge, and the LCP threshold correctly identified 2.6% of cancers for which the Brock model threshold would have recommended discharge, then using McNemar's test with 90% power and a significance

level of 0.05, 659 participants would be needed. However, with a sample size of 2000, there is 90% power to detect a difference between the methods if the percentages were as small as 1.0% and 2.3%, respectively.

▶ Number and percentage of benign-nodule patients discharged: again, results from thresholds defined for discharge in a previous study are used.[7] Based on previous data within the benign group, assuming for the discordant pairs that 17.0% of benign nodules are recommended for discharge by the LCP threshold but not by the Brock model threshold. In comparison, 13.5% are recommended for discharge by the Brock model threshold but not by the LCP threshold; there is 80% power to detect a significant difference between the two tests with 2026 individuals using McNemar's test. This equates to an OR of 1.26

Sample size calculations were performed using R statistical software (V.4.1.2; R Foundation for Statistical Computing, Vienna, Austria).

### Statistical analysis

Statisticians will carry out all analyses. For each outcome measure, the following will be reported:

▶ The number of participants included in the analysis, stratified by the relevant group.

▶ Mean or median (SD or IQR) for continuous outcomes, and numbers and proportions (percentage) for binary outcomes.

▶ An effect size (OR) with a 95% CI.

▶ A two-sided p value.

For both primary objectives, measured differences between standard care and the LCP options for patients with cancer diagnosed and benign-nodule patients discharged will be analysed using conditional logistic regression. Statistical significance will be assessed with a Wald test. Secondary outcomes will be described and analysed according to their distribution. All outcomes will have an exact p value provided for them so the reader can access the data for significance level, so correction for multiple testing should not be needed.

### Health economics analysis

The economic analysis will estimate the expected costs and expected quality-adjusted life years (QALYs) gained from standard care, LCP and LCP-guided care over patient lifetime from the perspective of the UK NHS. Incremental analyses will identify the most cost-effective permutation with exploratory analyses undertaken to determine the uncertainty around measures of cost-effectiveness. Costs will be calculated as quantities of resource use multiply by appropriate unit costs (eg, NHS Reference Costs) and will be specific to a year to be determined at time of analysis. A health economics analysis plan will be produced prior to undertaking the analysis at the end of the study patient follow-up period.

**Table 1** Study objectives and outcome measures

| Objective | Outcome measure |
|---|---|
| Primary: Determine the potential effect of the LCP on discharge | The measured difference between standard care and the three LCP-based options described under AI model for:<br>▶ No and percentage of cancer patients discharged (straight after assessment of the baseline scan).<br>▶ No and percentage of benign-nodule patients discharged (straight after assessment of the baseline scan). |
| Secondary: Determine the potential effect of the LCP on possible adherence to clinical guidelines | The measured difference between standard care and the three LCP-based options described under AI model:<br>▶ No and percentage of patients for whom a validated risk model (Brock or LCP) is used to guide the next clinical management step (ie, counting the instances where Brock is not used or where LCP is not possible to compute, or it is ignored). |
| Secondary: Determine the potential effect of the LCP on overall clinical management, as well as scan and procedure utilisation | No and percentage of cancer patients for whom there would have been a change in clinical management according to each of the three LCP-based options described under AI model compared with actual care (correctly for more aggressive management and incorrectly for less aggressive management).<br>No and percentage of benign-nodule patients for whom there would have been a change in clinical management according to each of the three LCP-based options described under AI model compared with actual care (incorrectly by more aggressive management and correctly for less aggressive management).<br>The measured difference between standard care and the three LCP-based options described under AI model for:<br>▶ No and percentage of CT scans and PET/CT scans performed on benign-nodule patients.<br>▶ No and percentage of non-surgical biopsies performed on benign-nodule patients.<br>▶ No and percentage of surgical excisions on benign-nodule patients. |
| Secondary: Determine the potential effect of LCP-guided care vs standard care on patient outcomes | The measured difference between standard care and the three LCP-based options described under AI model for:<br>▶ No and percentage of thoracic, respiratory or vascular events related to biopsies or surgical excisions for lung nodules or suspected lung cancer occurring within 30 days of the procedure on benign-nodule patients.<br>▶ No and percentage of lung cancers stratified by stage.<br>▶ Time in days between nodule detection and lung cancer diagnosis. |
| Secondary: Determine the potential health-economic effect of LCP-guided care versus standard care | The measured difference between standard care and the three LCP-based options described under AI model for:<br>▶ The composite standardised GBP costs of all healthcare-related activity for lung nodules or suspected lung cancer.<br>▶ Health-related quality of life (using EQ-5D-5L to generate QALYs).<br>▶ Estimate cost-effectiveness (Incremental costs per QALY gained). |

AI, artificial intelligence; LCP, lung cancer prediction; MDT, multi-disciplinary team; PET, positron emission tomography; QALY, quality-adjusted life year.

## Patient and public involvement

An extensive programme of patient and public involvement (PPI) input was arranged to provide input into the study objectives, design, materials and governance and planned to disseminate study results.

Specifically, a PPI representative, who sits on the Trial Management Group, provided input from study conception to developing study materials. This guidance and support will continue during the management and conduct of the study. An independent PPI contributor is a member of the trial steering committee. Further input was gathered from a group of PPI contributors selected from across the country with varying experiences of lung nodules, cancer or none from a patient and/or carer perspective during facilitated virtual workshops.

Input received to date has ranged from considerations of healthcare inequalities and other biases, awareness of AI generally and in healthcare, methods of and materials for participant recruitment and communication, and the design of any data collected during the study.

## ETHICS AND DISSEMINATION

This study has been reviewed and given a favourable opinion by the South Central—Oxford C Research Ethics Committee in UK (REC reference number: 22/SC/0142).

Given the study's observational nature, we do not envisage any elevation to the harms or the risk of harm for patients. The sponsor and chief investigator have conducted a full risk assessment.

Patients will be asked to give their consent to participate in the study (including the analysis of standard-care CT scans, associated data collection and the healthcare economics assessment) by telephone or face to face when attending routine clinic appointments. No additional patient visits, scans or procedures are planned for the study. Patients not wishing to participate in the health economics assessment may still participate in the study's CT imaging analysis.

The findings of this study will be shared with the funder (NHS Accelerated Access Collaborative). Results will be presented at academic and clinical conferences. The study will result in at least one peer-reviewed publication of the findings in an international medical journal.

**Author affiliations**
[1]Nottingham University Hospitals NHS Trust, Nottingham, UK
[2]King's College Hospital NHS Foundation Trust, London, UK
[3]Leeds Teaching Hospitals NHS Trust, Leeds, UK
[4]University of Nottingham, Nottingham, UK
[5]Optellum Ltd, Oxford, UK
[6]University Hospitals of Leicester NHS Trust, Leicester, UK
[7]Respiratory Medicine, St George's Hospital, London, UK
[8]Royal Brompton and Harefield Hospitals, London, UK
[9]Oxford University Hospitals NHS Foundation Trust, Oxford, UK
[10]University College London, London, UK
[11]Royal Marsden Hospital NHS Trust, London, UK
[12]Royal Free London NHS Foundation Trust, London, UK
[13]University College Hospital, London, UK
[14]Consumer Forum, NCRI CSG (lung) Subgroup, BTOG Steering Committee, NHSE CEG, National Cancer Research Institute, London, UK
[15]Population Health Sciences Institute, University of Newcastle, Newcastle upon Tyne, UK
[16]Royal Brompton and Harefield NHS Foundation Trust, London, UK
[17]University College London Hospitals NHS Foundation Trust, London, UK

**Contributors** All authors (EO'D, DRB, JW, FM, MB, MC, CVC, DC, ID, AD, JLG, FG, SJ, MK, RL, TMM, WM, JM, AN, JP, JR, GSS, AS, PS, RS, AT and RT) contributed to the development and set up of the study. EO'D, DRB, JW and FM wrote the manuscript, all authors (EO'D, DRB, JW, FM, MB, MC, CVC, DC, ID, AD, JLG, FG, SJ, MK, RL, TMM, WM, JM, AN, JP, JR, GSS, AS, PS, RS, AT and RT) contributed to, reviewed and approved the final version of the manuscript.

**Funding** This work is supported NHS Accelerated Access Collaborative (grant number AI_AWARD02266). Sponsor organisation is Nottingham University Hospitals NHS Trust.

**Disclaimer** The sponsor organisation and funder have had no role in study design.

**Competing interests** FG reports grant funding from NIHR and Innovate UK, consultancy fees from Polarean and has shares in Optellum; SJ has received grant funding from GRAIL, consultancy fees/ honoraria from AstraZeneca and Chiesi, stock options in Optellum Ltd and his spouse is employed by AstraZeneca; RL is funded by the Royal Marsden NIHR BRC, Royal Marsden Cancer charity and SBRI (including QURE.AI). RL's institution receives compensation for time spent in a secondment role for the lung health check program and as a National Specialty Lead for the National Institute of Health and Care Research. He has received research funding from CRUK, Innovate UK (cofunded by GE Healthcare, Roche and Optellum), SBRI, RM Partners Cancer Alliance and NIHR (coapplicant in grants with Optellum). He has received honoraria from CRUK; WM has received honoraria from Amgen; AN declares grants from Department of Health's NIHR Biomedical Research Centre's funding scheme and GRAIL, consulting fees from Aidence BV, Faculty Science LTD and MSD; Support for attending meetings from Takeda Limited; Advisory Board participation for Aidence BV and Faculty Science Ltd; JR has received support for travel and accommodation from BTOG, ERS, CRUK, EORTC, ESMO and NCRI; RT has received honoraria from Olympus Medical; DRB has grant from NHS Digital and had received honoraria from MSD and AstraZeneca; DC, FM, JW are Optellum employees and have share options in Optellum. The remaining authors declare no conflicts of interests.

**Patient and public involvement** Patients and/or the public were involved in the design, or conduct, or reporting, or dissemination plans of this research. Refer to the Methods section for further details.

**Patient consent for publication** Not applicable.

**Provenance and peer review** Not commissioned; externally peer reviewed.

**ORCID iDs**
Emma O'Dowd http://orcid.org/0000-0001-6904-1327
Tricia M McKeever http://orcid.org/0000-0003-0914-0416
Gurdeep Singh Sagoo http://orcid.org/0000-0003-1427-1437

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
