## [Reviewer comments · BMJ Open]

ARTICLE DETAILS

TITLE (PROVISIONAL)	Determining the impact of an artificial intelligence tool on the management of pulmonary nodules detected incidentally on computed tomography (DOLCE) study protocol: a prospective, non-interventional multi-centre United Kingdom study
AUTHORS	O'Dowd, Emma; Berovic, Marko; CALLISTER, Matthew; Chalitsios, Christos; Chopra, Disha; Das, Indrajeet; Draper, Adrian; Garner, Justin; Gleeson, Fergus; Janes, Sam; Kennedy, Martyn; Lee, Richard; Mauri, Fabrizio; McKeever, Tricia; McNulty, William; Murray, James; Nair, Arjun; Park, John; Rawlinson, Janette; Sagoo, Gurdeep; Scarsbrook, Andrew; Shah, Pallav; Sudhir, Rajini; Talwar, Ambika; Thakrar, Ricky; Watkins, Johnathan; Baldwin, David

VERSION 1 – REVIEW

REVIEWER	Schreuder, Anton Universiteit Leiden
REVIEW RETURNED	21-Jul-2023

GENERAL COMMENTS	Abstract: - a light recommendation to specify that the study focus is on "incidental" and "solid and part-solid" nodules Introduction: - Please include the missing reference for the 2015 BTS guideline on pulmonary nodules (page 3 line 27). Methods and analysis: - Unless the dates are currently unknown, please mention the expected start and end dates of recruitment/follow-up (page 4 line 12).- Is a one-year follow-up period sufficient to confidently diagnose all nodules, e.g., considering those which show slow growth but are not indicated for a biopsy (i.e., "probable benign-nodule patients")?- While it is written that the study focuses on incidental nodules, it would be good to explicitly describe this. I.e., that patients with pulmonary nodules detected on a CT which was prompted by suspicion of lung cancer will be excluded from the study.- Perhaps too detailed for the protocol (and possibly specified in the provided references), but is the AI risk score linear and calibrated? I.e., a value 2 nodule has double the malignancy risk of a value 1 nodule and half the malignancy risk of a value 4 nodule. And assuming that value 2 indicates 20% risk, are 20% of value 2 nodules malignant? It may otherwise be difficult for physicians to interpret each value.
--

	- Though it was reported earlier that this would be an observational study, it is described on page 5 that "The clinical team will have the option to modify management [...] after exposure to the LCP score. Since the management can be influenced by the risk scores, can this study still be considered observational? Perhaps an "observational-implementation hybrid" is a more suitable description of this study? - Will comorbidities and the management thereof be considered? I.e., considering situations where the risk score would typically change the nodule management but remains unchanged due to comorbidities. - When describing the outcome measures, it is a bit confusing when the term "standard care, LCP, and LCP-guided care" is used because it implies 3 options without clarification on what the difference between "LCP" and "LCP-guided" is. This should be corrected to "standard care and LCP-guided care" (if there are 2 options) or "standard care, LCP care, and LCP-guided care" (if there are 3 options). Another point of confusion is whether there will be a distinction between "LCP as an integer" and "LCP-CNN combined with a threshold-based rule-out recommendation" which are both to aid physician decision-making (i.e., are there actually 4 options?). In any case, the terms should be introduced earlier and be described explicitly (probably in the data collection subsection). The term "LCP-informed care" is also used at times; is this the same as "LCP-guided care"? - For the primary outcomes, which test will be used to calculate the p-value? Overall comments: Overall a well-written and timely study proposal to potentially endorse the implementation of AI to improve the nodule management workflow, reduce health care costs, and reduce radiologists' workload.
--	--

REVIEWER	Scherag, Andre Universitätsklinikum Jena
REVIEW RETURNED	31-Jul-2023

GENERAL COMMENTS	"Determining the impact of an artificial intelligence tool on the management of pulmonary nodules detected incidentally on computed tomography: DOLCE study protocol" is a study protocol for a multicenter observational study investigating if artificial intelligence (AI)-based malignant-nodule prediction solutions might outperform existing models. It is important to check such models scientifically prior to their wide-spread implementation in care. Major remarks:  • Please define by standard care, LCP, and LCP-guided care clearly in a special paragraph • Please state clearly which of the outcomes will be assessed blinded or more generally provide a figure showing at what point in time who has access to which information and what is documented by whom • The authors list two primary outcomes (as confirmatory test) – multiple testing in terms of type I and type II error control has to be addressed • Page 5; line 50-55: Should be attenuated – why not run a randomized trial – a diagnostic impact study? This should be the more ultimate (standard of care) changing test... • Page 6/7: exclusion criteria should not simply be the negation of the inclusion criteria (that would be redundant)
---

	 • Please provide information on the (standard care) grid of CT scans after baseline CT • Is individual consent necessary for this study and what bias will it likely introduce? • Has to meet the study protocol requirements to address the SPIRIT-AI Extension (https://www.equator-network.org/reporting-guidelines/spirit-artificial-intelligence/) where applicable Minor remarks:  • Page 5; line 23: Provide a reference for the statement • Page 5; line 38: Provide a reference for the validation statement • Page 6; line 7: How many centers are planned • Page 6; lines 17-24: Who is blinded – see also my major comment • Page 6; line 42: Define VNC • Page 8; line 42: If volumetry-based measures will be the standard in DOLCE, then ground truth 3b seems to be superfluous, right? • Page 10; line 30: Please provide details on the software used for sample size calculations
--	---

VERSION 1 – AUTHOR RESPONSE

REVIEWER 1

Comment	Response
Abstract: A light recommendation to specify that the study focus is on "incidental" and "solid and part-solid" nodules	We have added this to the Methods and analysis section of the Abstract
Introduction: - Please include the missing reference for the 2015 BTS guideline on pulmonary nodules (page 3 line 27).	We have now added this: Reference #5.
Methods and analysis:	
- Unless the dates are currently unknown, please mention the expected start and end dates of recruitment/follow-up (page 4 line 12).	Addressed as above with the following sentence: The start date was March 2023 with the planned recruitment end date being February 2024. With one year of follow up, this will mean a planned end date of February 2025.
- Is a one-year follow-up period sufficient to confidently diagnose all nodules, e.g., considering those which show slow growth but are not indicated for a biopsy (i.e., "probable benign-nodule patients")?	As per the 2015 BTS guidelines, one-year of follow up is sufficient for the definitions described under the 'Ground truth diagnosis' subsection, when determined through volumetry. For this reason, we have specifically excluded patients with part-solid nodules where the solid component is < 80% of the total nodule size, as these are more likely to represent slowly growing malignancies (and

	volumetry at one year is not sufficient to rule out malignancy).
- While it is written that the study focuses on incidental nodules, it would be good to explicitly describe this. I.e., that patients with pulmonary nodules detected on a CT which was prompted by suspicion of lung cancer will be excluded from the study.	Under the Study population subsection, we have now added the following phrase in bold: Patients who have undergone CT that has detected a pulmonary nodule and was performed for a reason other than suspicion of lung cancer will be reviewed for study eligibility (Table 1).
- Perhaps too detailed for the protocol (and possibly specified in the provided references), but is the AI risk score linear and calibrated? I.e., a value 2 nodule has double the malignancy risk of a value 1 nodule and half the malignancy risk of a value 4 nodule. And assuming that value 2 indicates 20% risk, are 20% of value 2 nodules malignant? It may otherwise be difficult for physicians to interpret each value.	While we agree with the Reviewer that this is an important point, we also agree that it is too detailed for the protocol. Moreover, the details behind the algorithm have been described in the validation papers of the Lung Cancer Prediction score (Baldwin et al., 2020 and Massion et al., 2020)
- Though it was reported earlier that this would be an observational study, it is described on page 5 that "The clinical team will have the option to modify management [...]" after exposure to the LCP score. Since the management can be influenced by the risk scores, can this study still be considered observational? Perhaps an "observational-implementation hybrid" is a more suitable description of this study?	We have incorporated the flexibility for actual-care management decisions to be modified only where there is a clear duty of care to the patient to do so, in light of new information. This new information may or may not be due to the LCP score. However, we anticipate such scenarios unfolding only in exceptional cases. In such cases, any change to actual care will be recorded for consideration during statistical analysis.
- Will comorbidities and the management thereof be considered? I.e., considering situations where the risk score would typically change the nodule management but remains unchanged due to comorbidities.	Yes – these will be considered and are part of the study's aim to assess the real-world clinical utility of the LCP score.
- When describing the outcome measures, it is a bit confusing when the term "standard care, LCP, and LCP-guided care" is used because it implies 3 options without clarification on what the difference between "LCP" and "LCP-guided" is. This should be corrected to "standard care and LCP-guided care" (if there are 2 options) or "standard care, LCP care, and LCP-guided care" (if there are 3 options). Another point of confusion is whether there will be a distinction between "LCP as an integer" and "LCP-CNN combined with a threshold-based rule-out recommendation" which are both to aid physician	Under the AI model subsection, we have now included some clarifications on the three LCP-based options. We have then referred to these three LCP-based options under: - Data collection subsection - The Outcome measures subsection These additions are highlighted in green. "LCP-informed care" is synonymous with "LCP-guided care".

decision-making (i.e., are there actually 4 options?). In any case, the terms should be introduced earlier and be described explicitly (probably in the data collection subsection). The term "LCP-informed care" is also used at times; is this the same as "LCP-guided care"? "	
- For the primary outcomes, which test will be used to calculate the p-value?	Wald tests will be used to calculate the p-value as part of the conditional logistic regression analysis. We have added a sentence to describe this.

REVIEWER 2

Major comments	
• Please define by standard care, LCP, and LCP-guided care clearly in a special paragraph	Under the AI model subsection, we have now included some clarifications on the three LCP-based options. We have then referred to these three LCP-based options under: - Data collection subsection - The Outcome measures subsection These additions are highlighted in green.
• Please state clearly which of the outcomes will be assessed blinded or more generally provide a figure showing at what point in time who has access to which information and what is documented by whom	Under Data collection , we have added the following sentence: At this point, the clinical team should remain blinded to the outcome of any actual-care investigation (e.g., PET/CT scan, biopsy etc.).
• The authors list two primary outcomes (as confirmatory test) – multiple testing in terms of type I and type II error control has to be addressed	We have added a sentence under Statistical analysis : Correction for multiple testing will be performed.
• Page 5; line 50-55: Should be attenuated – why not run a randomized trial – a diagnostic impact study? This should be the more ultimate (standard of care) changing test...	We agree with the reviewer that an RCT would be the ultimate study design. However, a sufficiently powered RCT would not have been possible within the funding and timeframe constraints.
• Page 6/7: exclusion criteria should not simply be the negation of the inclusion criteria (that would be redundant	The exclusion criteria are not just negation of the inclusion criteria. They highlight the fact that while there may be one or more qualifying nodules (new incidental 5-30mm solid / semi-solid), the patient may still be excluded due to some separate disqualifying nodules (e.g., nodule already being followed up)
• Please provide information on the (standard care) grid of CT scans after baseline CT	The number and timing of CT scans after the baseline CT scan will not be pre-specified within the protocol but will depend on local standard practice.

 • Is individual consent necessary for this study and what bias will it likely introduce? 	Yes, consent is needed as per UK regulations and as specified in the ethical review for this study. We don't envisage any bias being introduced. Several AI studies, including once using convolutional neural networks required individual consent and were successfully completed with no suggestion of bias and very high participant consent rates: [Baldwin DR, Gustafson J, Pickup L, Arteta C, Novotny P, Declerck J, Kadir T, Figueiras C, Sterba A, Exell A, Potesil V, Holland P, Spence H, Clubley A, O'Dowd E, Clark M, Ashford-Turner V, Callister ME, Gleeson FV. External validation of a convolutional neural network artificial intelligence tool to predict malignancy in pulmonary nodules. Thorax. 2020 Apr;75(4):306-312. doi: 10.1136/thoraxjnl-2019-214104. Epub 2020 Mar 5.]
 • Has to meet the study protocol requirements to address the SPIRIT-AI Extension (https://www.equator-network.org/reporting-guidelines/spirit-artificial-intelligence/) where applicable 	A sentence noting this has been added under Data collection. We have uploaded a version of the SPIRIT AI section with this revision.
Minor comments	
 • Page 5; line 23: Provide a reference for the statement 	Added.
 • Page 5; line 38: Provide a reference for the validation statement 	Added.
 • Page 6; line 7: How many centers are planned 	We have clarified throughout the text that 12 recruiting centres are planned.
 • Page 6; lines 17-24: Who is blinded – see also my major comment 	We have addressed this above.
 • Page 6; line 42: Define VNC 	We have defined VNC in the third paragraph of the Introduction
 • Page 8; line 42: If volumetry-based measures will be the standard in DOLCE, then ground truth 3b seems to be superfluous, right? 	Some of the sites will not use volumetry-based measure as part of their local standard care, so 3b is needed.
 • Page 10; line 30: Please provide details on the software used for sample size calculations 	We have added a statement regarding the software used under Sample size: Sample size calculations were performed using R statistical software (version 4.1.2; R Foundation for Statistical Computing, Vienna, Austria).

VERSION 2 – REVIEW

REVIEWER	Schreuder, Anton Universiteit Leiden
---

REVIEW RETURNED	10-Oct-2023
GENERAL COMMENTS	Thank you for adequately responding to my comments; I have no further comments and wish you the best of luck with your study.
REVIEWER	Scherag, Andre Universitätsklinikum Jena
REVIEW RETURNED	16-Oct-2023
GENERAL COMMENTS	Thanks for the changes; please specify how you plan to address multiple testing.

VERSION 2 – AUTHOR RESPONSE

REVIEWER 2

Comment	Response
Thanks for the changes; please specify how you plan to address multiple testing.	All outcomes will have an exact p-value provided for them so the reader can access the data for significance level, so correction for multiple testing should not be needed. If it is necessary to perform multiple testing, then we will use Bonferroni techniques. We have clarified this in the text.